**Data Availability Statement:** The data that support the findings of this study are available as supplementary appendix and on London School of

# Interventions to increase help-seeking for mental health care in low- and middle-income countries: A systematic review

Myrthe van den Broek[1,2‡], Yashi Gandhi[3‡], Diliniya Stanislaus Sureshkumar[4], Matthew Prina[5,6], Urvita Bhatia[3,7], Vikram Patel[8,9], Daisy R. Singla[10,11,12], Richard Velleman[3,13], Helen A. Weiss[14], Ankur Garg[3], Miriam Sequeira[3], Veera Pusdekar[3], Mark J. D. Jordans[1,2‡], Abhijit Nadkarni[3,15‡]*

1 Research and Development, War Child, Amsterdam, The Netherlands, 2 Amsterdam Institute for Social Science Research, University of Amsterdam, Amsterdam, The Netherlands, 3 Addictions and related-Research Group, Sangath, Goa, India, 4 Unit for Social and Community Psychiatry, Wolfson Institute of Population Health, Queen Mary University of London, London, United Kingdom, 5 Social Epidemiology Research Group, Kings College London, London, United Kingdom, 6 Population Health Sciences Institute, Faculty of Medical Sciences, Newcastle University, Newcastle Upon Tyne, United Kingdom, 7 Department of Psychology, Health and Professional Development at Oxford Brookes University, Oxford, United Kingdom, 8 Department of Global Health and Social Medicine, Harvard Medical School, Boston, Massachusetts, United States of America, 9 Department of Global Health and Population, Harvard T.H. Chan School of Public Health, Harvard University, Cambridge, Massachusetts, United States of America, 10 Campbell Family Mental Health Research Institute, Centre for Addiction and Mental Health, Toronto, Canada, 11 Department of Psychiatry, Temerty Faculty of Medicine, University of Toronto, Toronto, Canada, 12 Lunenfeld-Tanenbaum Research Institute, Sinai Health, Toronto, Canada, 13 Department of Psychology, University of Bath, Bath, United Kingdom, 14 MRC International Statistics and Epidemiology Group, London School of Hygiene & Tropical Medicine, London, United Kingdom, 15 Department of Population Health, London School of Hygiene & Tropical Medicine, London, United Kingdom

‡ MB and YG share first authorship on this work. MJDJ and AN are joint senior authors on this work.
* abhijit.nadkarni@lshtm.ac.uk

## Abstract

Mental health problems are a significant and growing cause of morbidity worldwide. Despite the availability of evidence-based interventions, most people experiencing mental health problems remain untreated. This treatment gap is particularly large in low- and middle-income countries (LMIC) and is due to both supply-side and demand-side barriers. The aim of this systematic review is to identify and synthesise the evidence on interventions to improve help-seeking for mental health problems in LMICs. The protocol was registered a priori (Registration number: CRD42021255635). We searched eight databases using terms based on three concepts: 'mental health/illness' AND 'help-seeking' AND 'LMICs'; and included all age groups and mental health problems. Forty-two papers were eligible and included in this review. Intervention components were grouped into three categories following the steps in the help-seeking process: (1) raising mental health awareness among the general population (e.g., distribution of printed or audio-visual materials), (2) identification of individuals experiencing mental health problems (e.g., community-level screening or detection), and (3) promoting help-seeking among people in need of mental health care (e.g., sending reminders). The majority of interventions (80%) included components in a combination of the aforementioned categories. Most studies report positive outcomes, yet results on

Hygiene and Tropical Medicine's Data Compass repository. It has been made available and can be cited using the following DOI: https://doi.org/10.17037/DATA.00003564.

**Funding:** This review was conducted as a part of a research study called IMPlementation of evidence-based facility and community interventions to reduce the treatment gap for depRESSion (IMPRESS) funded by a grant from National Institute of Mental Health [Grant number: RO1MH115504]. This grant supports Yashi Gandhi, Urvita Bhatia, Vikram Patel, Daisy Singla, Richard Velleman, Helen Weiss, Ankur Garg, Miriam Sequeira, Veera Pudsekar and Abhijit Nadkarni. Myrthe van den Broek and Mark Jordans were supported by Sint Antonius Stichting Projects (SAS-P) [Grant number: SAS-P-21103-UG]. Neither of the funders had any role in the study design and implementation of the research.

**Competing interests:** The authors have declared that no competing interests exist.

the effectiveness is mixed, with a clear trend in favour of interventions with components from more than one category. Ten out of 42 studies (24%) yielded a statistically significant effect of the intervention on help-seeking; and all targeted a combination of the aforementioned categories (i.e., raising awareness, identification and help-seeking promotion). Only six studies (14%) focused on children and adolescents. Due to the limited number of robust studies done in LMICs and the heterogeneity of study designs, outcomes and components used, no definite conclusions can be drawn with regards to the effects of individual strategies or content of the interventions.

## Introduction

Globally, over one billion people (15% of the world's population) were estimated to experience a mental or substance use disorder in 2019, accounting for 17% of years lived with disability from all causes [1]. Low- and middle-income countries (LMICs), where 80% of the global population lives [2], are at an increased disadvantage of experiencing mental health problems compared to high-income countries (HICs), since social factors such as poverty, inadequate housing, exposure to violence have a significant adverse effect on mental health, and the resources allocated to social and mental health care are limited [3]. Despite the burden, only a small fraction of those living with mental health problems in LMICs receive appropriate treatment–only 14% receive treatment, compared with 37% in HICs [4].

Many efforts have been made to overcome supply-side barriers (e.g., lack of trained professionals, limited financial resources) to mental health care by increasing access to, and availability of, care aiming to reduce the treatment gap for mental disorders [3]. These include: (1) the integration of mental health care into primary health care (PHC) [5]; (2) the use of digital technology to facilitate service delivery [6]; and (3) task-sharing with non-specialists [7]. While these are crucial innovations to scale mental health services in low-resource settings, uptake of these services is influenced by a complex interplay of supply-side and demand-side factors [3, 8, 9]. A lack of perceived need for treatment, attitudinal barriers such as social stigma, self-reliance, and perceived ineffectiveness of available care are frequently reported barriers to seeking help [10–14]. Hence, increasing the availability of mental health services alone will not necessarily lead to greater uptake of services [15].

Help-seeking in the context of mental health is defined as an attempt to obtain external assistance to deal with a mental health concern in a positive way [16]. A commonly used theory to predict human behaviour, including help-seeking, is the theory of planned behaviour. This theory describes how attitudes, subjective norms, and perceived control over the behaviour interact to influence intentions, and consequently the behaviour itself [17]. Accordingly, help-seeking interventions have focused on all three aspects of help-seeking i.e., attitudes, intentions, and behaviours, with often, attitudes and intentions becoming a proxy for actual help-seeking behaviour [16, 18]. However, the focus on attitudes and intentions may not be most efficient when targeting behaviour as the strength of associations between intentions and behaviour has not always been profound, leading to an 'intention-behaviour gap' [19]. Two recent systematic reviews [20, 21] found a wide range of interventions that focused on attitudes, intentions, and behaviours to seek help for mental health problems. Congruent with findings concerning the 'intention-behaviour gap' [19, 22], interventions that improved attitudes and intentions, did not necessarily lead to help-seeking behaviour [20]. Importantly, only the interventions that focused directly on engaging and motivating individuals experiencing a mental health problem improved behaviour [20]. Therefore, this systematic review

exclusively focuses on help-seeking behaviour. We apply a process model of help-seeking that conceptualises help-seeking as a dynamic multi-stepped process starting with creating awareness of symptoms and the need for help, to contacting and seeking external sources of help [23].

In addition, both previously conducted reviews only included trials, and none of these studies were conducted in low-income countries. Only three studies were based in the same middle-income country, China [21]. Considering that mental health care utilisation remains particularly low in LMICs [4] and that barriers contributing to this gap are context-driven and especially prominent in such settings, it is vital to identify contextual strategies to improve help-seeking. To build on the findings of the previously conducted reviews, this review included study designs beyond trials and also included regional databases. The aim of this systematic review is to identify and synthesise the evidence on interventions to improve help-seeking for mental health problems in LMICs.

The specific objectives of this review are to: (1) Synthesise the effectiveness of interventions in improving help-seeking behaviour, service utilisation and contact coverage; (2) Describe and categorise components and implementation processes of the interventions; (3) Synthesise information on factors that mediate help-seeking; and (4) Synthesise information about stakeholder perceptions and implementation-related outcomes of the interventions.

## Methods

This systematic review was conducted in accordance with PRISMA (Preferred Reporting Items for Systematic Reviews and Meta-Analyses) guidelines (see S1 Table) [24]. The protocol was registered a priori (Registration number: CRD42021255635; S1 Text).

### Search strategy

We searched eight databases between the 24th and 27th of May 2021, no publication date range were set in the databases: Medline, PsycINFO, Embase, Global Health, Cochrane Central, Latin American and Caribbean Health Sciences Literature (LILACS), Scientific Electronic Library Online (SCiELO), and African Journals Online (AJOL). Our search strategy was based on three main search concepts: 'mental health/illness' (e.g., anxiety, psychosis) *AND* 'help-seeking' (e.g., service use, contact coverage) *AND* 'LMICs' (e.g., low income nation). The detailed search strategy is attached as S2 Text. We also searched for potentially eligible studies in the reference lists of the two aforementioned systematic reviews [20, 21]. Additionally, we identified studies through forward and backward reference searching in Web of Science. Finally, four independent experts who are involved in help-seeking-related research in LMICs were contacted to identify additional published literature. We updated our results by re-running the search again between 9th and 16th January 2023 to account for studies that were published after May 2021 (when the search was first run).

### Eligibility criteria

Eligibility criteria were: (i) studies conducted in a LMIC, according to the World Bank categorisation [25]; (ii) peer-reviewed papers published in English; (iii) study participants of all ages, to distinguish it from previous reviews that focussed solely on children or adults; (iv) presentation of primary data; (v) evaluation of an intervention with a stated aim of promoting help-seeking behaviour; and (vi) reporting on at least one help-seeking behaviour related outcome (e.g., service utilisation or contact coverage). Studies evaluating the same intervention but conducted in another context or focusing on different outcomes were included as separate studies. Experimental, quasi-experimental and non-experimental studies were included.

## Study selection

After conducting the search, the results were imported to Endnote for automatic and manual de-duplication. The results were then uploaded to Covidence for title and abstract screening. Two researchers (MvdB and YG) independently screened the abstracts (n = 23,215); 8% were screened by both. In cases of disagreement, a third researcher (DSS) independently assessed eligibility. The remaining abstracts were screened by one of the researchers (MvdB and YG). Subsequently, each potentially eligible full-text article was independently assessed by two out of four researchers (MvdB, YG, DSS, or VeP). In cases where consensus was not reached, additional researchers (AN and MJ) determined eligibility. In the rare cases where the full version of the publication could not be found online, the corresponding author was contacted via email, and it was excluded if they did not respond within two weeks.

## Data extraction

Data was extracted on five key domains: general target study areas (country, mental health conditions, target population, setting); intervention (description, specific components, mode and agent of delivery, duration, frequency of sessions); primary outcome measures and results; hypothesised/studied mediators; and implementation-related outcomes. Two researchers (MvdB and YG) piloted the data extraction sheet by independently extracting information from 10% of the eligible papers. Data from the remaining papers were extracted by one of the researchers and reviewed by the other researcher. Some papers did not provide sufficient intervention details, and thus we accessed the paper the authors referenced for the intervention description. However, this was not included in our final list of included papers as it did not report on any relevant outcomes.

## Outcomes

The primary outcomes of interest were:

1. Help-seeking behaviour: any action taken to seek formal or informal help from mental healthcare services, or trusted individuals in the community [16].

2. Service utilisation: visiting any type of formal mental healthcare services [26].

3. Contact coverage: proportion of individuals in need of mental health treatment who seek help [8, 27].

4. Help-seeking efficacy: perceived helpfulness in promoting help-seeking [28].

Secondary outcomes of interest included hypothesised or studied mediators affecting help-seeking, implementation outcomes as defined by Proctor et al. [29] and, process indicators such as quality of training, session attendance, etc.

## Risk of bias

The risk of bias was assessed by two reviewers (MvdB and YG) using the Joanna Briggs Institute (JBI) critical appraisal tools relevant for the appropriate study design [30]. It was used to evaluate the methodological quality and the possibility of bias in its design, conduct and analysis. After independent evaluation, the two reviewers met to reach consensus on the score. All assessed papers were included in the review, irrespective of the scoring or quality.

## Analysis

Considering the heterogeneity in study designs and outcome measures, results were reported descriptively as a narrative synthesis [31]. Sub-groups were created based on study design, outcome measures and intervention content (i.e., intervention components). Intervention components were grouped into three categories following a process model of help-seeking that conceptualises help-seeking as a dynamic multi-stepped process [23]. This was summarised in a series of tables and graphical representations. Findings relating to effectiveness were summarised based on the intervention content. Implementation outcomes were presented based on common themes.

## Results

A total of 42 papers that represented 39 interventions across 18 LMICs were included (Fig 1). A total of 38,267 studies were identified by our search strategy; 15,152 were excluded as they were duplicates (n = 15,128) or were journals instead of individual papers (n = 24). The titles and abstracts of the remaining 23,215, were screened, 730 were eligible for full-text screening. Seventeen additional papers were identified from reference searching and ten were recommended by our independent experts. Forty-two papers were eligible and included in our review. There was agreement for 92.7% in the first abstract screening phase and the inter-rater reliability (IRR) for data extraction using the intraclass coefficient (ICC) was 0.79.

### Characteristics of studies

A summary of the 42 eligible studies (sample size range from 8–3,587 participants) and their characteristics is shown in Table 1. Studies were conducted between 1998 and 2022. Six were RCTs, seven quasi-experimental, five cohort, 20 cross-sectional and four case series. The studies were conducted in 19 countries in Asia (n = 31), Africa (n = 14), South America (n = 3), and North America (n = 2), with some being multi-country. Nearly 40% of the interventions targeted people experiencing mental health problems (n = 18), of which four focused on pregnant women. The other 60% focused on general population *and* people experiencing mental health problems, of which six focused on children, and adolescents.

### The interventions

**Intervention categories.**   Interventions were grouped into three categories following the broad steps in the help-seeking process: (1) raising mental health awareness (74% of the interventions included a component targeting this goal), (2) identification of individuals experiencing mental health problems (76%), and (3) promoting help-seeking among people identified as in need of care (64%). The majority of interventions (80%) included components in a combination of the aforementioned categories (see Table 2).

**Intervention content.**   Each intervention included a varying number of components (i.e., activities, elements or techniques described in the paper) (see Table 3). The mean number of components per intervention was four (ranging from 1 to 10 components).

For raising mental health **awareness**, 17 different intervention components were found. The primary aim was to increase knowledge about mental health problems and available services. Most components targeted the general population and included distribution of printed materials in the community such as posters and leaflets [34, 35] and social interactions and, house-to-house campaigns [36, 37].

Six intervention components were used for **identification** of individuals experiencing mental health problems. Most frequently reported components included community- or facility-

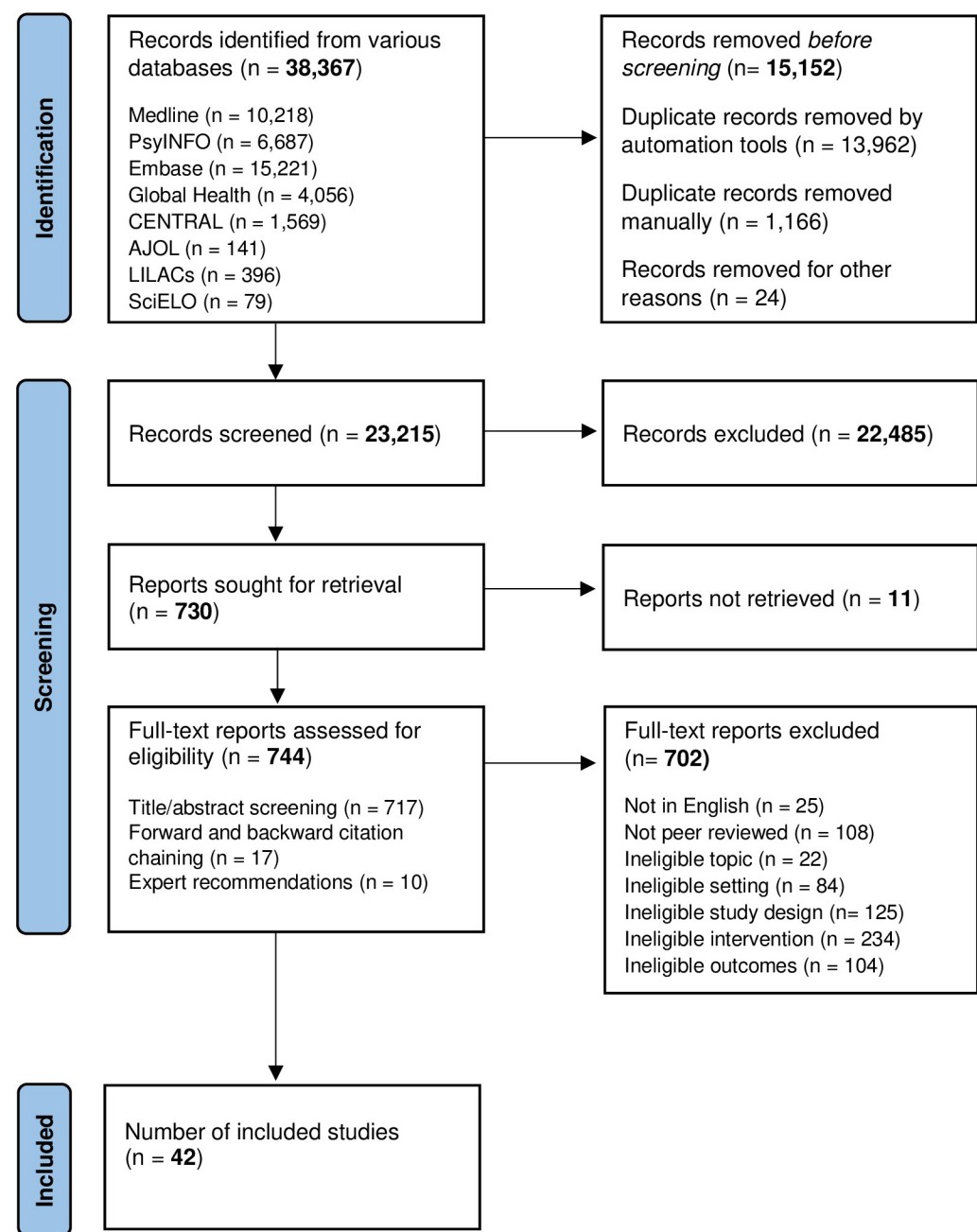

**Fig 1. Flow chart for selection of studies in the review.**

level systematic screening, case detection and online self-screening. Case detection activities were different from systematic screening as they were based on proactive identification by non-specialists [38, 39]. Furthermore, five studies used a tool developed to support proactive detection (i.e., the Community Informant Detection Tool [CIDT] [40]).

Nine **help-seeking promotion** components were identified and these were always coupled with an awareness raising and/or an identification activity. For example, sending text messages or calling people as a reminder where to find mental health care followed by messages to motivate patients to seek help [41, 42] or to provide a paper-based referral card [43] about

**Table 1. Summary of study characteristics, intervention(s) and reported outcome(s).**

| Author (Date of publication) | Year of Study | Study Design | Country | Target Population | Intervention | Outcomes |
|---|---|---|---|---|---|---|
| Bhardwaj et al. (2020) | 2017 | Case series | Nepal | People experiencing mental health problems | Proactive case detection and help-seeking promotion along with SMS functionalities to liaise among service providers. | 25% treatment seeking rate over three-months (i.e., 2/8 referred visited the facility). |
| Byaruhanga et al. (2008) | 2002–2004 | Cross-sectional | Uganda | General population | Awareness raising and setting up mental health outreach services for CMDs. | In total 12957 patients (2621 new and 10336 re-attending) were seen at 10 outreaches. A four-fold increase in the number of new patients over three years (from 388 to 1549) was observed. |
| Chavan et al. (2012) | 2004–2009 | Cross-sectional | India | People experiencing mental health problems | A helpline and awareness raising in tandem with other stakeholders. | 16.4% (183/1114) of the patients visited the OPD after advice. |
| Diez-Canseco et al. (2018) | 2015 | Case series | Peru | People experiencing mental health problems | A mental health screening app for PHCPs with web-based data collection and automated SMS text message delivery to motivate and remind patients. | 72.4% (92/127) of patients reported seeking specialized care on advice of a PHC provider; 55.1% (70/127) had at least one consultation in a specialised service after screening. |
| Eaton et al. (2008) | 2001–2004 | Cross-sectional | Nigeria | (i) General population; (ii) People experiencing mental health problems | Mental health awareness program which included training of VHWs in identification and referral. | Number of new patients seen was statistically significant in both locations (v2 = 149.82, df = 11 and p< 0.05; v2 = 683.89, df = 11 and p<0.05). |
| Eaton et al. (2017) | 2011–2012 | Quasi-experimental | Nigeria | (i) General population; (ii) People experiencing mental health problems | Mental health awareness program which included training of VHWs in identification and referral. | Incident rate of new patients for the initial period was 5.1 times higher than at baseline (95% CI 3.42–7.56, p<0.001). This diminished in the longer term. |
| Gaiha et al. (2021) | Not reported | Cross-sectional | India | General population | Mental health awareness campaign. | 1,176 persons attended over 20 one-day mental health screening camps after the awareness campaign. |
| Green et al. (2020) | 2019 | Case series | Kenya | Pregnant women experiencing mental health problems | AI to drive automated chats with patients and taking them through a psychological support intervention. | 66% (27/41) women sent at least one message to begin the registration process; 34% (14/41) of these engaged beyond registration. |
| Gong et al. (2020) | 2016–2017 | Cohort | China | Pregnant women experiencing mental health problems | Sending alert text messages by text with screening results and information about consultation for perinatal depression. | Only 1.2% (3/248) of the high-risk patients who were sent an initial alert reached out for further information. Out of the 3, only 1 attended their appointment. |
| Hailemariam et al. (2019) | Not reported | Cross-sectional | Ethiopia | People experiencing mental health problems | Training community workers to detect and refer people to PHCs where mental health care has been integrated. | Contact coverage of 81.3% (300/369). |

*(Continued)*

**Table 1.** (Continued)

| Author (Date of publication) | Year of Study | Study Design | Country | Target Population | Intervention | Outcomes |
|---|---|---|---|---|---|---|
| Hajebi et al. (2021) | 2015–2017 | Quasi-experimental | Iran | (i) General population; (ii) People experiencing mental health problems | National program to integrate mental health services into PHC. | Mental health service utilisation by general population was significantly higher (19.5 in case vs 13.1 in control, p < 0.001) and by patients with mental disorders was not significant (48.8 in control vs 43.1 in control districts, p = 0.19). |
| | | | | | | Inpatient mental health service utilisation before and after the intervention increased in case districts (0.9% vs 1%) and reduced in control (0.6% vs 0.4%). |
| He et al. (2020) | 2020 | Case series | China | (i) General population; (ii) People experiencing mental health problems | Awareness generation, hotline consultation, video consultation and on-site crisis intervention. | 4,236 accessed the hotline consultation; 233 received online video psychotherapy and 70 were provided with one-on-one crisis intervention. |
| James et al. (2002) | Not reported | Quasi-experimental | India, Pakistan | General population | Training PHC workers and providing integrated mental health care at the facility. | There was differential impact of interventions for some types of contact, but only in Pakistan. |
| Jordans et al. (2017) | 2014 | Cohort | Nepal | People experiencing mental health problems | Training lay health workers in proactive case detection and help-seeking promotion. | 67% (341/509) accessed a healthcare facility after being referred. |
| Jordans et al. (2019) | 2013–2017 | Cross-sectional | Nepal | (i) General population; (ii) People experiencing mental health problems | Community sensitization and training lay health workers in proactive case detection and help seeking promotion. | Contact coverage significantly increased for all disorders based on actual utilisation data of 12 months. The increases ranged from 7.5% for AUD to 50.2% for psychoses. |
| Jordans et al. (2020) | 2016–2017 | RCT | Nepal | People experiencing mental health problems | Training FCHW in proactive case detection and help-seeking promotion. | Median number of patients registered was 46.9% greater at facilities with CIDT training than at those with standard training (p = 0.04, r = 0.42). |
| Kaewanuchit et al. (2019) | 2017 | Cross-sectional | Thailand | General population | Policy to increase access to the health care system including mental health care. | The standardized regression weight of the community policy on service utilization was 0.050 (p<0.01; $R^2$ = 0.308). |
| Khan et al. (2017) | 2012–2013 | RCT | Pakistan | Pregnant women experiencing mental health problems | Brief interactive psychoeducation-based intervention for pregnant women. | 71% of women/their family contacted LHWs for assistance in the intervention arm as compared to 46% in control (p = 0.036). |
| Kutcher et al. (2016) | 2014–2015 | Quasi-experimental | Tanzania | (i) General population; (ii) People experiencing mental health problems | School-based mental health literacy program. | 63% of the teachers noted that they had personally sought help 6-months after participation in the program. |
| Kutcher et al. (2017a) | Not reported | Cohort | Tanzania | (i) General population; (ii) People experiencing mental health problems | School-based mental health literacy program. | Over a one-year period, 4657 students were exposed to the program (146 per teacher) and 399 (8.6%) of students approached a teacher regarding a mental health concern after the training program. |

(*Continued*)

**Table 1.** (Continued)

| Author (Date of publication) | Year of Study | Study Design | Country | Target Population | Intervention | Outcomes |
|---|---|---|---|---|---|---|
| Kutcher et al. (2017b) | Not reported | Cross-sectional | Malawi | (i) General population; (ii) People experiencing mental health problems | School-based mental health literacy program and enhancement of clinical competencies among community health care providers. | 122 adolescents visited the health clinic post referral from teachers. No denominator reported. |
| Lee et al. (2022) | 2022 | Cross-sectional | Myanmar | (i) General population; (ii) People experiencing mental health problems | Psychosocial support focal point programme supporting referral and linkages to more specialized services for internally displaced persons (IDPs). | 5,725 households received handouts and 679 visits were conducted to IDPs. From this, 332 (48.9%) resulted in a phone call to a counsellor. |
| Leykin et al. (2013) | 2005–2007 | RCT | India, South Africa, Argentina, Mexico | People experiencing mental health problems | Intervention involving receiving online prompt messages, post screening. | Receiving a stronger prompt predicted greater likelihood of seeking treatment (p<0.002, OR = 1.66, 95% CI: 1.20–2.30). |
| Liu et al. (2019) | 2017–2018 | Cohort | China | People experiencing mental health problems | Online proactive identification of suicide risk combined with specialized crisis management. | 24,727 direct messages to 12,486 different social media users were sent, and 5542 (44.4%) of them responded by completing the assessment protocol or by interacting with the counsellors. |
| Luitel et al. (2019) | 2013–2017 | Cross-sectional | Nepal | People experiencing mental health problems | Mass sensitisation, stigma reduction, and training lay health workers proactive case detection and help seeking promotion, as part of a multi-level mental health care plan. | The proportion of the participants receiving treatment for depression increased from 8.1% (18/228) at baseline to 11.8% (13/118) at follow up (RR = 1.40, p = 0.336), and for AUD from 5.1% (5/96) to 10.3% (9/74; RR = 2.33, p = 0.115). |
| Malakouti et al. (2022) | 2014–2015 | RCT | Iran | People experiencing mental health problems | A brief educational intervention and telephone contact (BIC) program to prevent suicide reattempts. | 36.4% of the participants from the BIC group reported a need for help during the follow-up period, of whom 74% used services (35/129). In the TAU group 2 out of 129 used these services. |
| Maulik et al. (2017) | 2014–2016 | Quasi-experimental | India | (i) General population; (ii) People experiencing mental health problems | Awareness raising campaign and tech-enabled service delivery model for screening, diagnosing and managing CMDs based on task sharing. | Mental health service utilisation increased from 0.8% (2/238) at baseline to 12.6% (29/232) at follow up. |
| Maulik et al. (2020) | 2014–2019 | Quasi-experimental | India | (i) General population; (ii) People experiencing mental health problems | Awareness raising campaign and tech-enabled service delivery model for screening, diagnosing and managing CMDs based on task sharing. | Self-reported use of primary care services for mental health problems significantly increased from 3.3% (30/900) at baseline to 81.2% (731/900) at follow up (95% CI 89.0 to 199.7; p<0.001). |
| Nakku et al. (2019) | 2013–2016 | Cross-sectional | Uganda | (i) General population; (ii) People experiencing mental health problems | Training VHW to raise awareness and identify people with mental health problems, as part of a multi-level program. | Contact coverage for depression increased from 16.5% (48/325) at baseline to 19.4% (94/452) at follow up (+4.1%; 95% CI – 1.8, 10.1; p = 0.173), and for AUD from 0.0 (0/25) at baseline to 1.3% (1/63) at follow up (+1.3%; 95% CI – 1.3, 3.9; p = 0.317). |

(*Continued*)

**Table 1.** (Continued)

| Author (Date of publication) | Year of Study | Study Design | Country | Target Population | Intervention | Outcomes |
|---|---|---|---|---|---|---|
| Nguyen et al. (2021) | 2016 | Cross-sectional | Vietnam | (i) General population; (ii) People experiencing mental health problems | Website for secondary school students providing information on mental health. | Nearly all students (98.6%) visited the website, 75% visited the website once or twice. 17.7% used it to search for help. |
| Parikh et al. (2021) | 2018 | SW-CRCT | India | (i) General population; (ii) People experiencing mental health problems | A classroom-based sensitisation program. | The proportion of students referred was significantly higher in the IC (IC = 21.7%, CC = 1.5%, OR = 111.36, 95% CI 35.56–348.77, p<0.001). The proportion of self-referred participants was also higher in the IC (IC = 98.1%, CC = 89.1%, Pearson $\chi^2$ (1) = 16.92, p<0.001). |
| Pradeep et al. (2014) | 2006–2009 | RCT | India | General population | Enhanced care provided by CHWs. | Participants in the enhanced care arm were 2 times more likely to visit the clinic as compared to treatment as usual. Number of visits were significantly greater in the intervention group (p<0.001). |
| Ragesh et al. (2020) | 2015–2016 | Cross-sectional | India | Pregnant women experiencing mental health problems | 24-hour telephonic helpline service for mothers discharged from a mother-baby psychiatry unit. | Among 113 mothers, 51 (45%) made 248 calls. 42% were made by the mothers and 48.3% by family members. |
| Rathod et al. (2018) | 2013–2014 | Cross-sectional | Ethiopia, India, Nepal, South Africa, Uganda | People experiencing mental health problems | Case detection by trained facility workers. | Treatment initiation among depression screen-positives was 0% in Ethiopia, India and South Africa, 0.5% in Nepal (1/179; 95% CI—0.0 to 3.9) and 4.2% in Uganda (2/48; 95% CI–1.0 to 15.4), and among AUD screen-positives was 0% in Ethiopia, India, South Africa and Uganda, and 2.2% in Nepal (2/90; 95% CI– 0.5 to 8.6). |
| Ravindran et al. (2018) | 2013–2016 | Quasi-experimental | Nicaragua | (i) General population; (ii) People experiencing mental health problems | School-based mental health literacy program. | Mean difference (0.86) in change in GHSQ scores in both groups pre- and post-curriculum was not significant (95% CI—0.48 to 2.20, p = 0.21). |
| Shaikh et al. (2016) | 2011–2015 | Cross-sectional | India | General population | Setting up community-led and peer-led support systems for transgender populations. | Access to mental health support increased from 43.7% at baseline to 77.2% at follow up (33%+; p<0.001). |
| Shidhaye et al. (2017) | 2013–2014 | Cross-sectional | India | People experiencing mental health problems | Awareness raising and to provide services through a collaborative care model. | The contact coverage for current depression was six-times higher in the 18-month survey population, from 4.3% (95% CI 1.5–7.1) at baseline to 27.2% (95% CI 21.4–33.7) at follow up (p<0.001). |
| Shidhaye et al. (2019) | 2014–2016 | Cross-sectional | India | (i) General population; (ii) People experiencing mental health problems | Awareness generation and detection at the facility, as part of a multi-level program. | Contact coverage at baseline was 14.8% (95% CI 11.1% to 19.4%) for depression and 10.5% (95% CI 7.6% to 14.3%) at follow up, and 7.7% (95% CI 2.8% to 19.5%) at baseline for AUD and 7.3% (95% CI 3.1% to 16.2%) at follow up. |

(*Continued*)

**Table 1.** (Continued)

| Author (Date of publication) | Year of Study | Study Design | Country | Target Population | Intervention | Outcomes |
|---|---|---|---|---|---|---|
| Shrivastava et al. (2012) | Not reported | Cross-sectional | India | People experiencing mental health problems | A community-based clinic with a crisis helpline for suicide prevention. | The helpline received 15,169 calls within five years, 15% of patients using the helpline accessed care from a mental health professional or psychiatrist. |
| Stein et al. (2001) | 1998 | Cross-sectional | South Africa | People experiencing mental health problems | Support group. | 64% reported "agree" and 25.5% reported "somewhat agree" to the role of the group in encouraging them to seek or continue help. |
| Tewari et al. (2017) | 2014–2016 | Cohort | India | (i) General population; (ii) People experiencing mental health problems | Awareness raising campaign and tech-enabled service delivery model for screening, diagnosing and managing CMDs based on task sharing. | 1,243 calls were placed to the community, ASHAs and doctors, 78.6% were heard by them. |
| Tzelios et al. (2022) | 2020–2021 | Cross-sectional | Peru | (i) General population; (ii) People experiencing mental health problems | Four mental health chatbots connecting people with a range of mental health services. | Services were provided to nearly all users of mental health chatbots who accepted to be contacted (42,932/42,933, 99.9%). |

AI = Artificial Intelligence; ASHA = Accredited Social Health Activists; AUD = Alcohol Use Disorder; CHW = Community Health Worker, CMD = Common Mental Disorder; FCHW = Female Community Health Worker; GHSQ = General Help-Seeking Questionnaire; LHW = Lay Health Worker; PHC = Primary Health Care; PHCP = Primary Health Care Provider; SW-CRCT = Stepped-wedge cluster randomised controlled trial; VHW = Village Health Worker.

Kutcher et al. (2017a) [32], Kutcher et al. (2017b) [33]

treatment options, mostly after a screening activity. Psycho-education was another common activity wherein a trained non-specialist worker, a community mental health worker (CMHW) for example, visited the house and helped the patient and their family better understand mental health problems, how to support the patient, and address any doubts concerning the treatment [35, 44]. Some interventions set up support groups with peers or other users of services (such as the 'consumer advocacy group' in South Africa) as an activity to enhance collective efficacy and help-seeking behaviour [45]. In some interventions with a help-seeking promotion component the identification component was either not described or included as a

**Table 2. Intervention categories following the steps in the help-seeking process.**

| Intervention categories | Number of studies (%) |
|---|---|
| Raising awareness (only) | 4 (11%) |
| Identification (only) | 4 (11%) |
| Help-seeking promotion (only) | 0 (0%) |
| Raising awareness, identification and help-seeking promotion | 14 (39%) |
| Raising awareness and identification | 4 (11%) |
| Raising awareness and help-seeking promotion | 4 (11%) |
| Identification and help-seeking promotion | 6 (17%) |

Note: One intervention is a policy reform with components outside of these categories and was not taken into account in this table.

**Table 3. Summary of intervention activities, elements or techniques.**

| Author(s) (year) | Community, stakeholder meetings | Public forums & press conferences | Social events | Types of performances | Word of mouth | Social media | Social contact/ interactions | Distribution of printed materials | Wall paintings | Audio/ audio-visual resources | Website | Interviews & news items in print | Radio/ TV interviews, announcement | Training/ workshop | Programs in schools/ colleges | Collaboration with stakeholders | Awareness raising–not specified |
|---|---|---|---|---|---|---|---|---|---|---|---|---|---|---|---|---|---|
| | | | | | | | | **Raising awareness** | | | | | | | | | |
| Kutcher et al. (2016) | | | | | | | | • | | | | | | | | | |
| Kutcher et al. (2017a) | | | | | | | | • | | | | | | | • | | |
| Kutcher et al. (2017b) | | | • | | | | | • | | | | | • | | • | | |
| Ravindran et al. (2018) | | | | | • | | | • | | | • | | | | • | | |
| Bhardwaj et al. (2020) | | | | | | | | | | | | | | | | | |
| Green et al. (2020) | | | | | | | | | | | | | | | | | |
| Hailemariam et al. (2019) | | | | | | | | | | | | | | | | | |
| James et al. (2002) | | | | | | | | | | | | | | | | | |
| Shrivastava et al. (2012) | | • | | | | | | | | | | | | | | | |
| Jordans et al. (2020) | | | | | | | | | | | | | | | | | • |
| He et al. (2020) | | | | | | | | | | | | | • | | | • | |
| Ragesh et al. (2020) | | | | | • | | | • | | | | | | | | | |
| Byaruhanga et al. (2008) | • | | | | | | | • | | | | | | | | | |
| Hajebi et al. (2021)[1] | | | | | | | | | | | | | | | | • | • |
| Jordans et al. (2019)[2] | • | | | | | | • | • | | • | | | | | | | |
| Luitel et al. (2019)[2] | • | | | | | | • | • | | • | | | | | | | |
| Shidhaye et al. (2017) | • | | | | • | | | | • | • | | | | | | | |
| Maulik et al. (2017) | | | • | • | • | | | • | | • | | | | | | | |
| Maulik et al. (2020) | | | • | • | • | | | • | | • | | | | | | | |
| Tewari et al. (2017) | | | • | • | • | | | • | | • | | | | | | | |
| Chavan et al. (2012) | | | | | | | | | | | | • | • | • | | • | |
| Shidhaye et al. (2019)[3] | • | | | | | | | • | | • | | | | | | • | |
| Eaton et al. (2008) | • | | | | | | | | | • | | • | • | | | | |
| Nakku et al. (2019)[4] | | • | | | | | | | | | | | | | | • | • |
| Eaton et al. (2017) | • | | | | • | | | • | | | | | • | | | | |
| Gaiha et al. (2021) | | • | • | • | | | • | • | | • | | | | | | • | |
| Khan et al. (2017) | | | | | | | | • | | | | | | | | | |

*(Continued)*

**Table 3.** (Continued)

| Author(s) (year) | Identification | | | | | | Help-seeking promotion | | | | | | | | | Total |
|---|---|---|---|---|---|---|---|---|---|---|---|---|---|---|---|---|
| | Screening at community/household | Screening at facility level | Detection with tool | Detection without tool | Self-disclosure | Automated/ algorithm screening | Psycho-education with patient | Psycho-education with family | Encouraging social support | Conducting home visits | Setting up support groups | Providing information via help-line | Sending digital prompts/ reminders | Provide first aid | Referral slip | |
| Rathod et al. (2018)[5] | | | | | | | • | • | | | | | | | | |
| Stein et al. (2001) | • | | | | | | | • | | | | | | • | | |
| Shaikh et al. (2016) | | | | | • | • | • | • | | | | | | • | | |
| Díez-Canseco et al. (2018) | | | | | | | | | | | | | | | | |
| Jordans et al. (2017) | | | | | | | | | | | | | | | | |
| Gong et al. (2020) | | | | | | | | | | | | | | | | |
| Leykin et al. (2013) | | | | | | | | | | | | | | | | |
| Liu et al. (2019) | | | | | | | | | | | | | | | | |
| Pradeep et al. (2014) | | | | | | | | | | | | | | | | |
| Kaewanuchit et al. (2019) | | | | | | | | | | | | | | | | |
| **Total** | 7 | 3 | 5 | 4 | 8 | 1 | 4 | 18 | 1 | 7 | 1 | 2 | 5 | 3 | 3 | |
| Kutcher et al. (2016) | | | | | | | | | | | | | | | | 1 |
| Kutcher et al. (2017a) | | | | | | | | | | | | | | | | 2 |
| Kutcher et al. (2017b) | | | | | | | | | | | | | | | | 4 |
| Ravindran et al. (2018) | | | | | | | | | | | | | | | | 4 |
| Bhardwaj et al. (2020) | | | • | | | | | | | | | | | | | 1 |
| Green et al. (2020) | | | | | | • | | | | | | | | | | 1 |
| Hailemariam et al. (2019) | • | | | | | | | | | | | | | | | 1 |
| James et al. (2002) | • | | | | | | | | | | | | | | | 1 |
| Shrivastava et al. (2012) | | | | | • | | | | • | | | • | | | | 3 |
| Jordans et al. (2020) | | | • | | | | • | • | | | | | | | | 4 |
| He et al. (2020) | | | | | • | | | | • | | | • | | | | 4 |
| Ragesh et al. (2020) | | | | | • | • | | | • | | | • | | | | 5 |
| Byaruhanga et al. (2008) | | • | | • | | | | | • | | • | | | | | 5 |
| Hajebi et al. (2021)[1] | | | • | | | | • | • | • | | • | | | | | 6 |
| Jordans et al. (2019)[2] | | | • | | | | | | • | | • | | | | | 6 |
| Luitel et al. (2019)[2] | | | • | | | | | | • | | • | | | | | 6 |
| Shidhaye et al. (2017) | | | | • | | | | | | | | | | • | | 6 |

*(Continued)*

**Table 3.** (Continued)

| Study | | | | | | | | | | | | | | Total |
|---|---|---|---|---|---|---|---|---|---|---|---|---|---|---|
| Maulik et al. (2017) | • | | | | | | | | | | | | • | 7 |
| Maulik et al. (2020) | • | | | | | | | | | | | | • | 7 |
| Tewari et al. (2017) | • | | | | | | | | | | | | • | 7 |
| Chavan et al. (2012) | | • | | • | | • | | • | | | • | • | | 9 |
| Shidhaye et al. (2019)[3] | • | | | • | • | | • | | • | | | | • | 10 |
| Eaton et al. (2008) | | • | | | | | | | | | | | | 4 |
| Nakku et al. (2019)[4] | | • | | | | | | | | | | | | 4 |
| Eaton et al. (2017) | | • | | | | | | | | | | | | 5 |
| Gaiha et al. (2021) | • | | | | | | | | | | | | | 8 |
| Khan et al. (2017) | | | | • | • | | | | | | • | | | 3 |
| Rathod et al. (2018)[5] | | | | • | • | | | | | | • | | | 4 |
| Stein et al. (2001) | | | | | | | • | | | • | | | | 5 |
| Shaikh et al. (2016) | | | | | | • | | | | • | | | | 6 |
| Diez-Canseco et al. (2018) | • | | | | | | | | | | • | | | 2 |
| Jordans et al. (2017) | | | • | | | | | | | | | | • | 2 |
| Gong et al. (2020) | • | | | | | | | | | | • | | | 2 |
| Leykin et al. (2013) | | | | • | | | | | | | • | | | 2 |
| Liu et al. (2019) | | | | | • | | | | | | • | | | 2 |
| Pradeep et al. (2014) | • | | | | | • | | | • | | | | • | 3 |
| Kaewanuchit et al. (2019) | | | | | | | | | | | | | | 0 |
| Total | 7 | 4 | 5 | 7 | 3 | 5 | 6 | 1 | 4 | 6 | 5 | 7 | 3 | 2 |

Note: Papers that were referenced and used in this review for the intervention description are:

[1] [48];

[2] [49];

[3] [50];

[4] [51];

[5] [52].

The colours indicate in which category the intervention components were grouped i.e., yellow is raising awareness, blue is identification, and orange is help-seeking promotion. Kutcher et al. (2017a) [32]; Kutcher et al. (2017b) [33]

study procedure. Furthermore, one-third of the interventions that included a help-seeking promotion component also included a follow up component such as home visits or phone calls to ensure adherence, overcome barriers and provide continuing psychosocial support to people with mental health problems [40, 46].

One intervention was a policy reform introducing public health administration for immigrant workers to increase access to mental health care but did not describe specific components in the above-mentioned categories [47].

Two-thirds of the help-seeking interventions (69%; n = 29) combined the above-mentioned demand-generating activities with a service provision intervention to improve access (see S2 Table). This included integrating mental health care into PHC or in schools, setting up community outreach clinics or 24-hour mental health helplines and offering online treatment using video or text messages with a mental health worker or chatbot. In addition, social support needs were also addressed by establishing links with other existing community resources, income generation opportunities and addressing the socio-economic factors that affect help-seeking [36].

## Implementation of the interventions

**Where: Type of setting.**   Over 70% of the interventions (n = 30) were implemented in-person; mainly in the community (n = 21), PHCs or community centres (n = 4), or in schools (n = 5). Others were delivered online (n = 5), through a chatbot [53, 54], a website [55], or via direct messages using social media [56, 57], via media advertisement (n = 3) [46, 58, 59] or automated text messages (n = 1) [42]. One provided both in-person information and reminders via the phone [60].

**Who: Delivery agent.**   For the interventions that were delivered in person, the most common delivery agents were non-specialist community health workers (CHWs) (n = 15) which consisted of members from formal health or affiliated systems, as employees or volunteers. This included health extension workers and village health workers (VHWs). Other delivery agents included non-state providers (NSPs) outside the health sector (e.g., teachers) (n = 4), NSPs in the health sector (e.g., PHCPs) (n = 2), mental health workers (e.g., psychiatric social worker) (n = 2), peer workers (n = 1) or a combination of these (n = 6). Peer workers included women, mothers, youth group members, or traditional healers, often selected based on criteria such as being motivated, respected and trusted in the community with basic literacy skills. In addition, five interventions were automated, either by sending alert messages with screening results [42, 56] or using machine learning to drive chats [53, 54, 57].

**What: Help-seeking source.**   The majority of interventions promoted help seeking through formal care, at a PHC centre (n = 23), community outreach clinics (n = 3), hospital (n = 1), school (n = 1) or a combination of these (n = 2) [39, 61]. The rest promoted remote mental health care via text messaging (n = 2) [53, 57] or a helpline (n = 6) [46, 58–60, 62, 63]. Two of these helplines were set up in response to mental health needs due to the COVID-19 pandemic [58, 63], for example through providing access to a program phone to connect with counsellors [63]. Two helplines combined counselling over the phone, with home visits or online video consultations based on individual needs [46, 58]. Two interventions promoted self-help through participation in support groups [45] or through learning certain skills to help cope with daily stress [55]. Collaboration with traditional healers in promoting help-seeking was mentioned in two studies [43, 64].

## Effectiveness of interventions to increase help-seeking

The results are organised by intervention category as presented in Table 2. The first two categories exclusively apply awareness raising or identification components, the remaining four

use multiple categories of intervention components. The results are further organised based on their effects: (1) positive, significant; (2) positive, not significant; (3) positive trend favouring the intervention but lacking data to evaluate effect sizes; and [4] no effect. Four help-seeking related outcomes were reported: mental health service-utilisation (n = 26 of 42; 62%; e.g., health centre visits, engagement with a service, or initiation of treatment); help-seeking behaviour (n = 8; 19%), contact coverage (n = 6; 14%) and help-seeking efficacy (n = 2; 5%).

**Raising awareness-only.** Four studies evaluated interventions that exclusively targeted awareness raising [32, 33, 44, 65]–all of which were cultural adaptations of the same school-based mental health literacy curriculum [66]. One trial [65] showed non-significant positive results in self-reported help-seeking behaviour pre- and post-curriculum. Two studies demonstrated a positive trend in help-seeking behaviour among students and teachers; 9% of the students reached out to teachers regarding a mental health concern [32], and 63% of the teachers personally sought help post-training [44]. Both combined the program with a supply-side component to integrate mental healthcare into PHC. The fourth study only reported a total number of 122 students seeking help after teacher referral of which 75% received a mental health-related diagnosis, however no denominator was reported [33].

**Identification-only.** Four studies evaluated interventions exclusively using identification components. One quasi-experimental study that evaluated a mental health training and support programme incorporated into PHC practice reported significant increase in service utilisation (p<0.05); however, this was true for both the intervention and control [67]. Two studies demonstrated a positive trend in help-seeking behaviour. A pre-pilot of an artificial intelligence (AI) driven mental health intervention for pregnant women in Kenya demonstrated that at least 66% of participants sent one message to register the application and half of those continued to engage beyond registration [53]. The other was a cross-sectional study in Ethiopia and reported a contact coverage of 81% (300 accessed care of 369 probable cases) after training CHWs in identification and referral [68]. Both interventions combined these demand-generating activities with a supply-side component. A case series in Nepal evaluating proactive case detection using the vignette-based tool (CIDT) combined with SMS functionalities to liaise among service providers found that only 25% (2 out of 8 referred) of the people identified sought treatment [69].

**Raising awareness, identification, and help-seeking promotion.** Sixteen studies evaluated a combination of intervention components targeting awareness raising, identification and help-seeking promotion; four showed a significant increase in service utilisation. The effectiveness of raising community awareness, proactive case detection and help-seeking encouragement using a vignette-based tool (CIDT) was compared to awareness raising only in an RCT in Nepal. The median number of patients registered was 47% greater after 6 months at facilities where proactive case detection was implemented (309 accessed care in the CIDT training arm vs 182 in standard training arm; p = 0.04, r = 0.42) [40]. A quasi-experimental study evaluated the SMART mental health project in India, which combined a help-seeking intervention comprising of an anti-stigma campaign, technology enabled screening and follow up messages for patients and a supply-side intervention of integrating mental health care into PHC. Self-reported service utilisation increased from 3.3% at baseline to 81.2% at follow up after 12 months of implementing the intervention (OR = 133.3, 95% CI 89.0 to 199.7; p<0.001) [70]. Two cross-sectional studies reported a significant increase in contact coverage. The first evaluated a community sensitisation program which included activities like wall paintings, detection by CHWs and provision of mental health first aid to encourage help-seeking [61]. Contact coverage increased six times over 18 months (from 4.3% at baseline to 27.2% at follow up; p<0.001). The second evaluated community sensitisation, proactive case detection using the CIDT and help-seeking encouragement [71]. Contact coverage based on service utilisation

data over 12 months increased from 0% at baseline to 7.5% at follow up for AUD, from 0 to 12.2% for depression, from 1.3 to 11.7% for epilepsy and from 3.2 to 50.2% for psychosis. Both interventions combined the demand-generating activities with mental health care integrated into PHC. A cross-sectional study evaluating proactive case detection using the CIDT combined with mass sensitisation, stigma reduction, and help-seeking encouragement showed a non-significant increase in contact coverage [72]. Contact coverage for depression increased from 8.1% at baseline to 11.8% at follow up and from 5.1% at baseline to 10.3% for AUD. A quasi-experimental study of a national program to integrate mental health care into PHC showed that provision of mental health screenings led to more people from the general population utilising mental health services and being diagnosed. Utilisation after the intervention was significantly higher in the case (19.5) than control districts (13.1) (p<0.000). However, among patients with mental disorders there was no significant increase in utilisation [73].

Eight studies showed a positive trend towards the intervention, however, did not provide data to evaluate effect sizes. Two evaluated the SMART mental health program and both showed an increase in service use [37, 74]. Four studies evaluated helplines that provided remote mental healthcare or referral to other services [46, 58, 59, 62] and three reported an increase in service utilisation [58, 59, 62]. One showed that over 70% contacted the help-line, however, only 16% sought help after being advised to do so [46]. One RCT evaluated a brief educational and telephone contact program that aimed to encourage individuals who had attempted suicide to seek help. In the intervention group 27% (n = 35) used the service, compared to 2% (n = 2) in the TAU group [60]. A chatbot introduced during COVID-19 was able to connect nearly all users who accepted to be contacted to care (42,932/42,933; 99.9%) [54].

**Raising awareness and identification.** Six studies evaluated the combination of awareness raising and identification [34, 38, 75, 76], and three showed a significant increase in help-seeking [75, 76]. One stepped-wedge cluster RCT evaluated the added impact of a classroom sensitization session compared to whole-school sensitization activities on demand for a school counselling service for adolescents [77]. The proportion of students referred in the former was significantly higher (21.7% vs 1.5%, OR = 111.36, 95% CI 35.56 to 348.77, p<0.001) [77]. The proportion of (self-) referred students was also higher in the classroom sensitization arm [77]. Two evaluated a mental health program in Nigeria which included awareness raising using local media and posters along with training of VHWs in identification and referral of people experiencing mental health problems to services integrated into PHC. A quasi-experimental study [76] showed a five times higher incidence rate of new patients, sustained for over a year after the intervention. The cross-sectional study [75] showed that the increase occurred with a strong temporal relationship to the training of VHWs and increases were mainly sustained for the following month, with a tail off from the very high levels immediately following the training. Meanwhile, another cross-sectional study in Uganda with similar intervention components, but specifically targeting depression and alcohol use disorder (AUD) found a non-significant increase in contact coverage (ranging from 1.3 to 4.1%) [38]. A cross-sectional study in India studied the impact of a mental health awareness campaign targeting the general public. It comprised of disseminating educational materials, conducting public meetings, street plays and quizzes, and organising screening camps. Nearly 66% received a preliminary diagnosis, indicating the success of campaign messages in improving recognition of mental health problems [34]. Finally, a cross-sectional study evaluating a website launched for secondary school students with information on mental health and psychological wellbeing showed that nearly all of the students (98.6%) visited the website [55] and 10–20% also used it for other reasons like seeking help and sharing information.

**Raising awareness and help-seeking promotion.** Five studies evaluated a combination of intervention components targeting awareness raising and help-seeking promotion. Only one

showed a significant increase in number of patients initiating treatment. This cross-sectional study evaluated the use of educational materials and a community-led and peer-led support system (e.g., a closed Facebook page) combined with providing facility-based one-on-one counselling, specifically for the transgender population in India. Access to mental health support increased from 43.7% at baseline to 77.2% at end line (p<0.001) [36]. Another cross-sectional study evaluated a focal point programme providing linkages to mental healthcare via a project phone during COVID-19, and 332 (48.9%) of the 679 visits resulted in a phone call to a counsellor. A cross-sectional study about a mental health support group in South Africa found that 64% of the participants rated the group to be helpful in encouraging help-seeking [45]. A feasibility trial in Pakistan evaluated psychoeducation by a lady health worker (LHW) at home and resulted in 71% of women or their family members contacting a LHW for assistance within 2 months of intervention completion, compared to 46% in the control arm (p = 0.036) [35]. Finally, a cross-country evaluation of case detection by trained facility workers in India, Nepal, Ethiopia, South Africa and Uganda found low to no change in treatment initiation for those who screened positive for depression or AUD [78].

**Identification and help-seeking promotion.** Six studies evaluated interventions that combined identification with help-seeking promotion, out of which two RCTs reported a significant increase in help-seeking behaviour. One trial compared the effectiveness of enhanced care by CHWs with regular referrals. Enhanced care included regular home visits and psychoeducation with the target population and their family members. Participants in the enhanced care arm were two times more likely to visit the clinic as compared to treatment as usual and a significantly higher number of visits were found in the enhanced care arm (p<0.001) [79]. A cohort study in Nepal, using a similar community-based approach, evaluated proactive case detection using the CIDT in combination with help-seeking encouragement by key community members [43]. About 67% of the people who were referred sought help, however no data was presented to evaluate effect sizes. The second RCT evaluated online self-screening and automated help-seeking promotion messages. They assessed the effect of intensity of prompts in the messages on help-seeking behavior [56]. This study supported the use of a stronger, more direct prompt message post screening rather than a 'lighter' one. Across all six countries, 10.2% reported seeking treatment after the light prompt at follow up and 16.6% following the strong prompt (p<0.002). A similar approach of sending help-seeking promotion messages was used in three other studies, out of which two demonstrated a promising trend in help-seeking behaviour. One combined a mental health screening app used by PHCPs with automated motivational text messages; over 70% of the patients receiving the message sought help [41]. Another cohort study evaluated identification of suicide risk using a machine learning model combined with automated messages with help-seeking options. About 44.4% of the people responded [57]. A cohort study evaluating the effects of help-seeking promotion text messages for perinatal depression in China, showed that only 1.2% reached out for further information and only one-third of them attended an appointment after receiving information [42].

One intervention was a policy reform for immigrant workers to increase access to mental health care [47]. The intervention components fell outside of the above created categories. But, a path model analysis found that introducing public health administration and the community policy reform had the most direct impact on mental health service utilisation (p<0.01).

**Reported implementation outcomes.** Eleven studies assessed implementation outcomes. The findings were primarily based on the perspectives of help-seekers, their caregivers, and CMHWs. In a few studies, key community stakeholders, mental health professionals and medical doctors also shared their experiences.

**Acceptability-related outcomes.** Eight studies reported on acceptability of the intervention [34, 35, 41, 53, 57, 62, 69, 74]. Acceptability was defined as the perception among implementation stakeholders that a given service or innovation is agreeable and palatable [29].

*Raising awareness.* Two studies reported high levels of acceptability, especially for raising awareness via public meetings, street plays and screening/health camps [34, 74]. Regular public meetings, for example, were considered to be like support groups for caregivers. Involvement of community leaders, accompaniment of a familiar person or a mental health expert at community events and, the use of culturally relevant illustrations in informational materials were considered as key facilitators [34, 35, 74].

*Identification.* People were willing to open up about their problems and listen to advice regarding help-seeking from CMHWs [41]. CMHWs are familiar with the context and well-respected in the community, they serve as gatekeepers to identify people at risk [62, 69, 74]. However, on the contrary, one study reported that CMHWs were unable to identify people; the reasons of which have been explained in the feasibility section [69]. Training for CMHWs was considered vital to build knowledge about mental health problems, develop basic counselling skills (e.g., rapport building), learn to identify people with mental health needs (with or without screening tools) and hone personal skills (e.g., building confidence) [69, 74].

*Help-seeking promotion.* The one study that assessed openness to digital therapy interventions found that most participants thought they could trust the app, perceived it to be unbiased and said that it helped them learn key skills which improved their health and relationships with others [53]. Additionally, around 70% of participants found proactive help through the use of direct messaging acceptable [57] and 90 to 95% of participants found a helpline service to be useful and said that they would recommend it to others [62].

**Feasibility and barriers to feasibility of implementation.** Five studies reported on feasibility and outlined key barriers [41, 53, 62, 69, 74]. Feasibility is defined as the extent to which an innovation can be successfully used or carried out within a given agency or setting [29, 80].

Two studies that assessed the feasibility of running an awareness campaign in Nepal and India found that there were high levels of stigma resulting in labelling and discrimination within the community [69, 74]. The stigma was also found to be prevalent among CMHWs who were found to be uncomfortable interacting with the potential AUD patients fearing they would be violent [69]. Another barrier outlined was related to convincing and psycho-educating family members to be supportive during treatment [62, 69, 74]. As a result, there were low levels of identification and help-seeking in these communities.

The CMHWs in Nepal also shared that they were not very confident using technology, citing reasons such as low levels of digital literacy and eyesight [69]. However, a study in Peru reported no such challenges [41]. Poor network was another challenge that people faced in rural areas [69, 74]. Additionally, maintaining patient confidentiality was a concern while employing mobile phones for screening and communication [69, 74].

Two studies that assessed experiences of CMHWs found that participants shared concerns with being overburdened due to multiple responsibilities in the health sector [41, 69]. A lack of financial incentives was also brought up and it was suggested that this would be an effective strategy to increase motivation for them to engage in the mental health sector [69].

Lastly, supply-side barriers including lack of availability of professionals for referral and unavailability of specialists were also highlighted [41, 62, 69, 74].

**Fidelity.** Two studies assessed fidelity; both reported satisfactory levels of fidelity on intervention delivery [35, 65]. Specifically, one reported achieving over 85% fidelity in curriculum delivery as determined by the project team's ratings [65].

**Appropriateness.** Two studies assessed on usefulness of the information circulated via different mediums like handouts, announcements on the loud speaker/radio and making it

available on a website [55, 63]. For instance, nearly 30% report using skills from the "stress and coping" and "changing unhelpful thoughts" handout on a daily basis in their lives and about 65% report using them a few times a week [63]. Moreover, nearly 80% of the respondents found it "very useful" when the handouts were played on the radio and/or loudspeakers [63] and nearly 75% reported the website to be very useful [55]. The students also reported that some of the topics covered by the website such as stress, depression, substance abuse, were very relevant to them [55]. However, the health check-up section which focussed on ways to measure stress, depression and self-esteem were rated appropriate by only 19% of the students [55].

**Reported moderating variables.**   No studies reported on factors that mediate help-seeking. However, a couple of studies explored variables that influenced help-seeking. Being in debt [61], longer distance to the nearest health centre [43], gender (being women) [38] and higher levels of functional impairment [68] were associated with increased levels of help-seeking behaviour. Whereas the same variables of symptom severity [53] and longer distance [68] were most commonly reported barriers in other studies. Additionally, being pregnant (vs. new mother), working out of the home [53], being widowed, divorced, or separated [61] was associated with lower levels of help-seeking behaviour.

A study that explored the use of a website to provide information and increase help-seeking found that boys found the website more appealing and were also more likely to share information [55].

**Risk of bias.**   The studies were generally of mixed quality except for RCTs and quasi-experimental studies, which proportionally received least negative appraisals on the items. The mixed score was predominantly due to a high level of unclarity or lack of details, and often, a 'not applicable' score, which has been discussed in the limitations. For cross sectional and cohort studies, reporting on confounding factors and strategies to deal with those was rarely found. For cross sectional, the information regarding the objectivity of the outcome measure and how the measurement was conducted was in most studies unclear. For case series studies, the research sites (e.g., prevalence or details about the population) were mostly not described sufficiently and there was a lack of information on statistical analysis. See S3 Table for assessment of all studies.

## Discussion

This systematic review identified 42 papers reporting 39 interventions across 19 LMICs. The field of help-seeking intervention research is rapidly emerging: nearly 60% of the papers were published within the past five years. The majority of interventions (80%) targeted a combination of three categories following the steps in the help-seeking process: raising mental health awareness, identification of individuals experiencing mental health problems, and promoting help-seeking among people in need of mental health care. Most studies report positive outcomes, yet results on the effectiveness of interventions is mixed, with a trend in favour of interventions with components from more than one category.

### Trends in intervention content and effectiveness

Ten studies (24%) showed a statistically significant increase in help-seeking behaviour [36, 40, 56, 61, 70, 71, 75–77, 79]. This is encouraging given the few trials found in LMICs in previous systematic reviews [20, 21, 81]. Help-seeking intervention research often use two main (separate) approaches; universal interventions aimed at the general population targeting attitudes and intentions, and indicated interventions targeted at those who experience mental health problems [20, 81]. Previous reviews found that interventions that improved attitudes and

intentions, did not necessarily lead to improved help-seeking behaviour [20]; in line with this, none of the interventions which only focused on information sharing activities were found to improve help-seeking behaviour [81]. We found a similar pattern for interventions that exclusively used awareness raising components for example. Although the studies showed a positive trend in help-seeking behaviour, the results were not significant or lacked data to evaluate effectiveness. Hence, to have a population level effect on help-seeking, a combined approach was recommended in the prior review [20]. The current review shows a similar trend; all ten studies that yielded a statistically significant effect on help-seeking behaviour at post intervention targeted multiple categories.

Out of the ten studies with positive significant findings, all but one (i.e., an online intervention) [56], involved non-specialists like CHWs and peer workers as the primary delivery agents. Five studies took place in India, two in Nepal and two in Nigeria. Delivery agents were often selected as trusted and engaged individuals, with regular contact with community members or students. This strategy may be especially beneficial in areas where mental health literacy is low, and where formal health services are not necessarily seen as a place to seek treatment for mental health problems. Furthermore, community members use shared cultural idioms which could promote more effective communication, facilitate trust and therefore lower barriers to seeking help and ultimately, reduce stigma [82, 83]. These in-person interventions all linked people to mental healthcare based in the community, and one in schools [77]. This integration of mental healthcare into community contexts like PHC follows recommended models to tackle supply-side barriers such as accessibility and costs of services [3, 84].

Two RCTs demonstrated that a low-intensity intervention using the task-sharing approach resulted in significantly more number of visits to mental health care. Jordans and colleagues [40] evaluated a vignette-based tool to support proactive community level detection by key community members in Nepal; the median number of patients registered was 47% greater after 6 months. Pradeep and colleagues [79] found that CMHWs in India could effectively monitor and psycho-educate patients to improve help-seeking. Participants were two times more likely to visit the clinic as compared to treatment as usual. From a public health perspective, these results are promising especially given the low-intensity of the interventions and the use of the task-sharing approach. These results are consistent with findings that showed that psychological treatments using a limited number elements were effectively delivered by non-specialist providers with moderate to strong effects sizes [83].

Two quasi-experimental studies with statistically significant results showed an increase in help-seeking behaviour of 5 [76] to 23 times higher [70] over the course of one year. The intervention with the greatest impact (i.e., the SMART mental health project in India) also appears to be of highest intensity. The intervention focused on strengthening collaboration between implementation agents and service providers and tailored pre-recorded messages were sent to screen-positive individuals, Accredited Social Health Activists (ASHAs) and doctors [70]. However, this study also reported that only 33% of the cases identified by CMHWs were clinically diagnosed with a mental health problem. Although this did not pose an issue in this study since there was sufficient availability of services and could be explained by natural remission, in other settings this could pose a potential risk of overburdening the health system. Following the hypothesised importance of the perceived need for mental health care as a mediator for actual help-seeking [85], further optimising the accuracy of identification may boost effectiveness of help-seeking interventions. Another important caveat is that only Eaton et al. [76] assessed long-term impact and showed that the effect of an mental health awareness program in Nigeria gradually tailed off after one year. Similar challenges in sustaining the results (e.g., drop in detection or app use) was highlighted by other studies [38, 53]. This may especially pose a challenge for interventions that apply task-sharing approaches or work with community

members as it often results in increased work pressure and challenges in sustaining motivation [86]. Recommended ways to improve motivation and continued engagement are substantial investments in all layers of staff, regular supervision meetings and, structured refresher trainings [39, 75, 76]. However, the reliance of most help-seeking interventions on this group warrants further investigation on sustainability of interventions from their perspective.

Another element that appears to be promising is the use of new technology to send digital prompts (e.g., text or online messages). Two tech-enabled interventions showed a significant increase in help-seeking behaviour. One is the SMART mental health project described above, and included sending pre-recorded messages to screen-positive individuals, ASHAs and doctors. This active follow up resulted in adherence of nearly 100% [70]. The other intervention was evaluated in six countries, including India, South Africa, Argentina and Mexico and showed that stronger help-seeking promotion messages generally resulted in greater likelihood of seeking care [56]. This is especially promising as increasingly, more people have access to a digital device and search for health information online [6]. Previous systematic reviews, mainly in HICs, found mixed results regarding the use of new technology. Gulliver et al. [81] for example, found two interventions conducted online (i.e., via email and a website plus telephone contact) both of which successfully increased formal help-seeking while Xu et al. [21] found no effectiveness for internet-based interventions and only short-term effects for telephone-based interventions.

## Target population

Given the influences of social networks on help-seeking behaviour, the interventions in our review not only focused on individuals with mental health problems, but also on important gatekeepers, such as family members or teachers, who can recognise symptoms and 'open the gate' to support [87].

Six studies (14%) focused on children and adolescents out of which only one reported a significant increase in help-seeking [77]. This classroom sensitisation program in India was delivered by a lay counsellor and used short animated videos with moderated group discussions. During the time periods when the classroom sessions were delivered, the referral proportion rose significantly. According to this and a previously conducted systematic review [21]—that found no effect for interventions targeting children—this is the only help-seeking promotion intervention for adolescent mental healthcare yielding a statistically significant effect on help-seeking. Two-thirds of the students that sought help in this trial did not meet the clinical thresholds for symptom severity, functional impact and chronicity, and further research may therefore be needed to avoid over-detection. Furthermore, school-based programs like these will need to be coupled with school-counselling services and suitable interventions for adolescents who do not meet clinical thresholds.

Studies also showed different impact of the same intervention based on the mental health problem that was targeted or the gender of the target population. For example, contact coverage increased by 8% for AUD but 50% for psychosis in a study in Nepal [71], and another study showed that adult men were not adequately targeted by the intervention implemented by VHWs in Uganda [38]. Different strategies may therefore be beneficial based on the target group.

## Implications and future directions

Although this review shows a promising trend for the effectiveness of strategies increasing help-seeking behaviour, there has been a limited number of robust studies undertaken in LMICs. This review has highlighted that especially interventions which used a combination of

intervention components resulted a statistically significant effect on help-seeking behaviour. It is therefore recommended that future interventions include various activities targeting awareness-raising among the general population, identification of individuals experiencing a mental health problem and specific help-seeking promotion. Furthermore, research on long-term effects of the intervention as well as sustainability (in terms of maintaining motivation and providing compensation to delivery agents) needs to be further explored. Moreover, it should be acknowledged that even if effective help-seeking interventions exist and utilisation increases, this does not guarantee a positive effect on mental health outcomes. A combination of demand (e.g., user-level factors) and supply-side factors (e.g., facility-, provider- level factors) need to be studied to understand the success in clinical effects. In other words, we need to develop a better understanding on how the help-seeking process works, and which combination of demand- and supply-side interventions are most effective in LMICs. Some studies pointed out to the importance of culture in the help-seeking process. For example, one study explained low uptake by placing the emphasis on joint decision making for a family member experiencing mental health problems [42]. Areas for further exploration also include involvement of family members and people with lived experiences, which is a central active ingredient in interventions to reduce stigma, enhance the demand for services [88] and on interventions focussing on children and adolescents.

## Limitations

The review only included papers published in English. Due to the heterogeneity in outcome measures and study designs, only limited conclusions on the effectiveness of strategies could be drawn. Furthermore, although the risk of bias was assessed, the quality of each study did not impact the weight given to it in the narrative synthesis. Moreover, the intervention activities were extracted and included only if they were explicitly reported in the publication. Lastly, the IRR for the abstract screening phase was not assessed.

## Conclusions

Overall, the present findings show a promising increase in attention to help-seeking intervention research in LMICs. This review highlighted those interventions which used a combination of intervention components targeting a combination of three categories following the steps in the help-seeking process: awareness raising, identification and help-seeking promotion appear to be the most effective in increasing help-seeking behaviour. However, due to the limited number of robust studies done in LMICs and the heterogeneity of study designs, outcomes and activities used no definite conclusions can be drawn with regards to the effects of individual strategies or content of the interventions.

## Supporting information

**S1 Table. PRISMA checklist.**
(DOCX)

**S2 Table. Supply-side components.**
(DOCX)

**S3 Table. Risk of bias results.**
(DOCX)

**S1 Text. Registered PROSPERO protocol.**
(PDF)

**S2 Text. Search strategy (Medline).**
(DOCX)

## Acknowledgments

We thank Dr. Gabriela Koppenol-Gonzalez, Senior Researcher at War Child, for her help with conducting statistical tests for inter-rater reliability. This review was conducted as a part of a research study called IMPlementation of evidence-based facility and community interventions to reduce the treatment gap for depRESSion (IMPRESS). The grant was awarded by National Institute of Mental Health to Sangath [Grant number: RO1MH115504]. War Child was supported by a foundation that wishes to remain anonymous. The foundation had no role in the study design and implementation.

## Author Contributions

**Conceptualization:** Myrthe van den Broek, Yashi Gandhi, Diliniya Stanislaus Sureshkumar, Matthew Prina, Mark J. D. Jordans, Abhijit Nadkarni.

**Data curation:** Myrthe van den Broek, Yashi Gandhi, Diliniya Stanislaus Sureshkumar, Veera Pusdekar.

**Formal analysis:** Myrthe van den Broek, Yashi Gandhi.

**Funding acquisition:** Abhijit Nadkarni.

**Methodology:** Myrthe van den Broek, Yashi Gandhi, Diliniya Stanislaus Sureshkumar, Matthew Prina, Mark J. D. Jordans, Abhijit Nadkarni.

**Supervision:** Matthew Prina, Vikram Patel, Mark J. D. Jordans, Abhijit Nadkarni.

**Writing – original draft:** Myrthe van den Broek, Yashi Gandhi.

**Writing – review & editing:** Myrthe van den Broek, Yashi Gandhi, Diliniya Stanislaus Sureshkumar, Matthew Prina, Urvita Bhatia, Vikram Patel, Daisy R. Singla, Richard Velleman, Helen A. Weiss, Ankur Garg, Miriam Sequeira, Veera Pusdekar, Mark J. D. Jordans, Abhijit Nadkarni.

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
