## [Decision Letter · Decision Letter 0]

13 Jun 2023

PGPH-D-22-01985

Interventions to Increase Help-seeking for Mental Health Care in Low-and Middle-Income Countries: A Systematic Review

Dear Dr. Nadkarni

Thank you for submitting your manuscript to PLOS Global Public Health. After careful consideration, we feel that it has merit but does not fully meet PLOS Global Public Health’s publication criteria as it currently stands. Therefore, we invite you to submit a revised version of the manuscript that addresses the points raised during the review process.

We look forward to receiving your revised manuscript.

Kind regards,

Darshini Govindasamy

Academic Editor

Journal Requirements:

2. Please provide separate figure files in .tif or .eps format only and remove any figures embedded in your manuscript file. Please also ensure all files are under our size limit of 10MB.

3. We notice that your supplementary tables are uploaded with the file type 'Figure'. Please amend the file type to 'Supporting Information'. Please ensure that each Supporting Information file has a legend listed in the manuscript after the references list.

4. In the online submission form, you indicated that "The data that support the findings of this study are available from the corresponding author upon request". All PLOS journals now require all data underlying the findings described in their manuscript to be freely available to other researchers, either 1. In a public repository, 2. Within the manuscript itself, or 3. Uploaded as supplementary information.

Additional Editor Comments (if provided):

We thank the authors for conducting this review. It highlights the need for more evaluations of mental health interventions in LMICs.

Reviewers' comments:

Reviewer's Responses to Questions

**Comments to the Author**

1. Does this manuscript meet PLOS Global Public Health’s publication criteria? Is the manuscript technically sound, and do the data support the conclusions? The manuscript must describe methodologically and ethically rigorous research with conclusions that are appropriately drawn based on the data presented.

Reviewer #1: Yes

Reviewer #2: Yes

2. Has the statistical analysis been performed appropriately and rigorously?

Reviewer #1: Yes

Reviewer #2: N/A

3. Have the authors made all data underlying the findings in their manuscript fully available (please refer to the Data Availability Statement at the start of the manuscript PDF file)?

Reviewer #1: Yes

Reviewer #2: Yes

4. Is the manuscript presented in an intelligible fashion and written in standard English?

Reviewer #1: Yes

Reviewer #2: Yes

5. Review Comments to the Author

Reviewer #1: Abstract:

Line 13: This treatment gap is particularly large in in low- and middle-income countries (LMIC) and is due to both supply-side and demand-side barriers.

Repetition of the word "in"

Introduction:

The authors provide sufficient background to an important public health issue - mental health/illness, particularly in the LMIC context and highlight disparities between LMICs and HICs in addressing the problem. They outline the barriers to access and uptake of mental health services in LMICs and highlight the need for scalable innovations to reduce the treatment gap.

In their rationale for the SR, the authors describe the theory underpinning the majority of help-seeking interventions, and outline the contributions of previous work to the body of knowledge, which helps to highlight the gaps they intend to fill and importantly, they acknowledge the need for contextual strategies to improve help-seeking in LMICs.

Line 43: Many efforts have been made to overcome supply-side barriers….Can you name some of these supply side barriers to provide the reader more context?

The aims and objectives of the SR are clearly defined.

Methods:

The search strategy was appropriate and the process rigorous. Eligibility criteria is more inclusive in comparison to previous reviews. However, the authors chose to include studies published in English only, which is noted later in the manuscript as a limitation of the study.

Line 94: concepts: ‘mental health/illness’ (e.g., anxiety, psychos?s).

Typo on spelling of psychosis.

Results:

The authors provide a comprehensive summary of the included studies and describe the various interventions which have been grouped into three intervention categories following the steps in the help-seeking process. A summary of how the interventions were implemented is presented and outlines the who, what and where. The effectiveness of the interventions is presented according to the intervention categories previously mentioned and various combinations of two or all of intervention categories. Implementation outcomes, feasibility (including barriers to implementation), appropriateness are also presented.

Discussion and Conclusion:

The discussion is well aligned with the aim stated in the introduction which was to identify and synthesise the evidence on interventions to improve help-seeking for mental health problems in LMICs. The authors have summarised the key findings and interpreted the results appropriately. I would have liked to see some of the LMIC countries or continents of included studies mentioned in the discussion for context. The authors note the contributions of this important piece of work to this topic and acknowledge some important limitations of the study. Implications for future research are also noted. The final conclusions made are supported by the data.

The acronym ASHA for Accredited Social Health Activists is only written in full in the footnote of Table 1. Please write in full at first mention in the manuscript text (Line 551?).

Reviewer #2: Article overview

In this study, the authors sort to identify interventions to improve help-seeking for mental health problems in LMICs. To investigate possible causes of the gap existing between mental health detection and linkage to care/treatment, the authors conducted a systematic review for evidence of interventions that actively sought to bridge this gap. Specifically, their synthesis focused on interventions aimed at improving help-seeking for mental health problems in LMICs. Although not framed in a formal question, the authors state that their review aimed to identify and synthesise the evidence on interventions to improve help-seeking for mental health problems in LMICs. The authors identified 42 articles that are relevant to the topic under investigation. They report that interventions using a combination of interventions to target health-seeking showed the most improvement in help-seeking behaviour and eventual linkage to mental health therapy/treatment. However, due to the limited number of robust studies done in LMICs and the heterogeneity of study designs, outcomes and activities used, the authors could not reach any definite conclusions regarding the effects of individual strategies or content of the interventions.

Reviewer comments to author

The article is well presented and easy to follow. It touches on a very important topic as most people in LMICs are either unaware of mental health problems (or what mental health entails) or are do not know that help is available.

I do, however, have the following minor points that need clarification:

1. Line 151:

The authors show that they conducted risk of bias analysis using JBI in supplemental documentation (S4). However, the scoring (or how subsequent decisions were made based on the JBI appraisal) is not evident (e.g. score ranges and weighting, were there any cut-offs suggested to guide decisions around bias, what was deemed acceptable vs not, etc.).

2. Lines 176-177

The authors state: “Half of the interventions targeted people experiencing mental health problems (n=18)”. Is this a reference to another total (i.e. n=36), because earlier it is reported that 42 eligible studies were identified (line 172). If so, then perhaps this sentence can be rephrased so as to prevent any confusion.

3. Lines 275-276

Minor comment. The numbering of “effects” is similar to reference citing in the manuscript. Perhaps using 1 bracket instead of 2 could avoid possible confusion [ i.e. 1) instead of (1)].

4. General comments on Tables and Figures

Figure 1: In screening, under “Full-text reports assessed for Eligibility”, it is suggested that the authors state the total number of articles before detailing how they went about assessing eligibility. Adding totals will maintain consistency with other sections in the diagram.

In addition, the authors might have explained how they moved from “Reports sought for retrieval=730” to “Number of included studies=42” in the body on the manuscript. However, this does not come across clearly in the diagram since 702 reports were excluded from the 730.

Tables 1-3: Although captions are clear enough for reader to understand what is being communicated, they, however, seem too general. It is suggested that authors be a bit more descriptive to enhance captions.

6. PLOS authors have the option to publish the peer review history of their article (what does this mean?). If published, this will include your full peer review and any attached files.

**Do you want your identity to be public for this peer review?** For information about this choice, including consent withdrawal, please see our Privacy Policy.

Reviewer #1: No

Reviewer #2: No

---

## [Editor Report · Decision Letter 1]

25 Jul 2023

Interventions to Increase Help-seeking for Mental Health Care in Low-and Middle-Income Countries: A Systematic Review

PGPH-D-22-01985R1

Dear Dr. Nadkarni,

We are pleased to inform you that your manuscript 'Interventions to Increase Help-seeking for Mental Health Care in Low-and Middle-Income Countries: A Systematic Review' has been provisionally accepted for publication in PLOS Global Public Health.

Best regards,

Darshini Govindasamy

Academic Editor